# Diversity of Monofloral Honey Based on the Antimicrobial and Antioxidant Potential

**DOI:** 10.3390/antibiotics11050595

**Published:** 2022-04-28

**Authors:** Anca Hulea, Diana Obiștioiu, Ileana Cocan, Ersilia Alexa, Monica Negrea, Alina-Georgeta Neacșu, Călin Hulea, Corina Pascu, Luminita Costinar, Ionica Iancu, Emil Tîrziu, Viorel Herman

**Affiliations:** 1Faculty of Veterinary Medicine, Banat University of Agricultural Sciences and Veterinary Medicine “King Michael I of Romania” Timisoara, Calea Aradului No. 119, 300645 Timisoara, Romania; anca.hulea@usab-tm.ro (A.H.); calin.hulea@usab-tm.ro (C.H.); corinapascu@usab-tm.ro (C.P.); luminita.costinar@usab-tm.ro (L.C.); ionica.iancu@usab-tm.ro (I.I.); emiltirziu@usab-tm.ro (E.T.); viorelherman@usab-tm.ro (V.H.); 2Faculty of Food Engineering, Banat University of Agricultural Sciences and Veterinary Medicine “King Michael I of Romania” Timisoara, Calea Aradului No. 119, 300645 Timisoara, Romania; ersiliaalexa@usab-tm.ro (E.A.); monicanegrea@usab-tm.ro (M.N.); 3Faculty of Agriculture, Banat University of Agricultural Sciences and Veterinary Medicine “King Michael I of Romania” Timisoara, Calea Aradului No. 119, 300641 Timisoara, Romania; alinaneacsu@usab-tm.ro

**Keywords:** honey, antimicrobial activity, antioxidants, manuka honey, acacia honey, linden honey, brassica rapeseed honey

## Abstract

This study aimed to investigate the antioxidant profile and the antimicrobial activity of four different types of monofloral honey (manuka (MH), brassica rapeseed (BH), acacia (AH), and linden honey (LH)) against some bacterial/fungal ATCC strains and some multidrug-resistant strains isolated from chronic otitis in dogs. For the characterisation of the antioxidant profile of each honey, we extracted the honey samples by hydroalcoholic extraction and analysed them in terms of total polyphenols (TPC), total flavonoids (TFC), and 2,2-diphenyl-1-picrylhydrazyl (DPPH) using the spectrophotometric method. The antimicrobial activity was determined using the microdilution method at concentrations of 10%, 15%, and 20%, with the results expressed in OD (optical density) calculated as BIR% (bacterial inhibition rate)/MIR% (mycelial inhibition rate). The antioxidant characterisation of the analysed honey samples showed the highest antioxidant activity and concentrations of TPC and TFC in MH, followed by LH. MH was proven to be the most effective on most clinical isolates concerning the antimicrobial activity in comparison with BH, AH, and LH. Except for *B. cepacia* and *P. vulgaris*, all the clinical isolates were sensitive to the antibacterial activity of honey. Regarding the ATCC strains, MH 10% was the most effective in inhibiting all the strains tested except for *P. aeruginosa.* In conclusion, the efficacy classification in our study was MH > BH > AH > LH.

## 1. Introduction

The increasing emergence of multidrug-resistant bacteria has become a significant challenge worldwide for veterinary and public health [1,2,3,4]. Repeated exposure to many antibiotics to treat various infections has resulted in bacterial resistance. The clinical evidence for this situation is represented by the increased incidence of chronic or recurrent bacterial disease [5]. According to studies, honey can be an alternative or complementary antimicrobial strategy with the advantage that, unlike traditional antibiotics, bacterial resistance to honey has not been reported [6,7,8].

For centuries, honey has been a curative treatment for various pathologies in traditional medicine due to its complex chemical composition with anti-inflammatory, antimicrobial, antiparasitic, antifungal, and antiviral properties [9,10,11,12]. The biological activity of honey, namely, its chemical composition, varies depending on the botanical origin (type of honey); the geographical origin; the meteorological conditions; and last but not least, the harvesting, processing, and storing conditions [13,14,15,16]. Depending on these factors, honey can be a potent antimicrobial agent whose bacteriostatic and bactericidal actions are comparable to antibiotics [14]. Several components contribute to the antimicrobial activity of honey, such as sugar content, endogenous hydrogen peroxide, polyphenol compounds, 1,2-dicarbonyl compounds, and bee defensin-1. Their synergic action allows honey to be active against various microorganisms, including multidrug-resistant bacteria, and modulates their resistance to antimicrobial agents [17]. In addition, manuka honey, monofloral honey obtained from *Leptospermum scoparium*, contains high concentrations of methylglyoxal (MGO), an organic compound with an antimicrobial activity whose concentration defines the unique manuka factor [18].

The mechanism of the antibacterial activity of honey has not been entirely elucidated [19]. The results of numerous studies about the broad spectrum of antimicrobial activity of honey against both Gram-positive and Gram-negative organisms are contradictory. Some studies claim that the antimicrobial activity of different types of honey shows a more significant inhibitory activity to Gram-positive organisms, such as methicillin-resistant *S. aureus* and *Staphylococcus epidermidis*, as well as reduced effect on Gram-negative bacteria such as *Pseudomonas aeruginosa*, *Enterobacter* spp., and *Klebsiella pneumoniae* [20,21,22,23,24,25]. The difference between the two types of Gram bacteria to the susceptibility of honey may be explained by the composition of the cell wall, with Gram-positive bacteria having, in addition to Gram-negative bacteria, an outer membrane protecting the peptidoglycan layer [24]. On the other hand, other studies have identified that Gram-positive bacteria are more resistant to honey than Gram-negative ones. Observations of Rubus honey from Southwest Spain showed *Proteus mirabilis* was the most susceptible organism tested, exhibiting MIC values ranging from 7.8 to 31.3 mg/mL, while *S. aureus* had MIC values ranging from 125 mg/mL [26]. Similar results of Egyptian honey identified that the only effective honey against *S. aureus* was Sidr honey (*Ziziphus Spina-Christi*) at a MIC of 100%, and only four out of six honey samples were effective against *Streptococcus mutans*.

All honey samples tested were effective against *P. mirabilis* and *K. pneumoniae* at MIC values of 50% or less [27]. The variety of results suggests that the antimicrobial activity of all kinds of honey is largely variable. However, studies have shown that dark honey had better antimicrobial activity than light honey due to the high concentration of polyphenols [28].

The biological activity of manuka honey also includes antifungal activity, demonstrated at concentrations of 0.01–70% (*v*/*v*) against *Candida albicans*, *Rhodotorula* spp., *Aspergillus niger*, *Epidermophyton floccosum*, *Microsporum gypseum*, *Trichophyton rubrum*, and *Trichophyton mentagrophytes* [29,30,31].

Although many studies have shown the antimicrobial efficacy of different kinds of honey, the therapeutic effects were mainly for skin infections, even for multidrug-resistant strains [7,32,33,34]. However, only a few data refer to the effectiveness of honey in treating other localisation of infections [35,36,37,38,39]. Canine infectious otitis externa (OE) is a pathological condition caused by various microorganisms that frequently become recurrent. Due to this fact, to treat OE, it is necessary to use multiple antibiotics for a long period of time, with the repeated therapy causing the emergence of antimicrobial resistance. Thus, an alternative treatment must be considered. So far, mainly manuka honey has been tested for the treatment of acute [37] and recurrent eczematous external otitis [38], having antimicrobial activity against *S. pseudintermedius*, *Klebsiella pneumoniae*, *Enterococcus cloacae*, *P. aeruginosa*, and *Malassezia pachydermatis otic isolates* [39]. 

The present research design was set as a preliminary in vitro study to highlight the antioxidant and antimicrobial activity of different types of honey against various microorganisms involved in OE so that in the future, in pet clinics, honey could be an alternative to the treatment of this condition. The activity of honey depends on the botanical origin and geographical area; therefore, this study highlighted the antioxidant activity, as polyphenols are responsible for the antimicrobial activity of four types of monofloral kinds of honey, three of them collected in western Romania. Alongside commercial manuka honey (Australian honey—*Leptospermum scoparium*), the three locally produced types of honey, which are easy to find at affordable prices compared to manuka honey, were acacia honey (*Robinia pseudoacacia*), linden honey (*Tilia* spp.), and rapeseed honey (*Brassica napus*). In the second phase of the study, ATTC strains and isolates from dogs with chronic otitis were studied to characterise the antimicrobial profiles. 

## 2. Results

### 2.1. Antioxidant Profile

#### 2.1.1. Total Polyphenol Content

The content of total polyphenols in the studied samples is shown in Figure 1. 

The mean values of TPC (*p* < 0.05) varied within fairly wide limits, from 177.320 to 528.160 mg GAE/kg sample. The highest TPC value was recorded for the honey sample MH (528.160 ± 3.043 mg GAE/kg sample), followed by LH (274.467 ± 6.150 mg GAE/kg sample) and BH (232.120 ± 8.620 mg GAE/kg sample), with the lowest value was recorded for the AH honey sample (177.32 ± 2.022 mg GAE/kg sample). The applied t-test showed a significant difference in the total flavonoid content between the samples analysed, except for BH and AH, where there were no differences. 

#### 2.1.2. Total Flavonoids Content (TFC)

The total flavonoid content (TFC) of the analysed samples is presented in Figure 2. 

Mean TFC values (*p* < 0.05) were within the limits of 21.490 and 65.563 mg/kg sample. The highest TFC value was recorded for MH (65.563 ± 0.664 mg/kg sample), followed by LH (51.380 ± 0.450 mg/kg sample) and BH (32.980 ± 0.395 mg/kg sample), with the lowest was recorded for AH (21.490 ± 0.320 mg/kg sample). Linearity similar to that recorded for the total polyphenol content was also obtained when the total flavonoid content was determined. The t-test showed a significant difference in the total flavonoid content between the samples analysed, except for BH and AH, where there were no differences.

#### 2.1.3. Antioxidant Capacity by 1,1-Diphenyl-2-picrylhydrazyl (DPPH) Assay

The antioxidant activity (*p* < 0.05) was found to be between 47.927 and 68.247%, as shown in Figure 3.

The antioxidant activity followed the same trend as the TCP and TCF. The highest antioxidant activity was recorded for MH (68.247 ± 0.06%), followed by LH (54.638 ± 0.086%) and BH (49.576 ± 0.071%), whereas for AH, the lowest value was recorded (47.917 ± 0.098%). The differences between the types of honey were determined using the t-test. Accordingly, significant differences (*p* < 0.05) were found between all types of honey, except for BH and AH, for which no differences were recorded.

### 2.2. Antimicrobial Activity

To interpret the results of the antimicrobial testing, we calculated two indicators: BGR/MGR and BIR/MIR, using the Formulas (2) and (3) presented in Section 4.2.

Figure 4, Figure 5, Figure 6, Figure 7, Figure 8, Figure 9, Figure 10 and Figure 11 show the bacterial inhibition rate (BIR %)/mycelial inhibition rate (MIR %), calculated according to Formula (3), while Table 1 and Table 2 present the optical density (OD) reading of honey samples tested on the ATTC and isolates strains.

A table containing the BGR%/MGR% values (2) when different concentrations of honey samples were applied to the screened strains is presented in the Appendix A.

Comparing the BIR percentages, the most sensitive ATCC strains in the case of the MH samples were *H. influenzae, E. coli*, and *S. pyogenes* (Figure 4). For *H. influenzae,* the bacterial inhibition rate (BIR%), depending on the concentration tested (MH 10%, MH 15%, and MH 20%), varied from 29.78% to 46.07% (Figure 4). In the case of *E. coli,* MH proved a bacterial inhibition rate (BIR%) ranging from 3.96% to 33.84% compared to the negative control, and against *S. pyogenes,* BIR ranged from 20.01% up to 32.27%. MH showed a medium effect on *S. aureus*, *S. typhimurium*, and *C. albicans*, with values up to 20% inhibition at the highest concentration tested and low inhibitory effects against *S. flexneri* and *C. parapsilopsis* with inhibitory values ranging up to 11%. 

MH also proved to have a potentiating activity on *P. aeruginosa*. The negative values of BIR showed a synergistic activity of MH with the *P. aeruginosa* strain tested, with the obtained values proving a strain boosting effect demonstrated by a bacterial mass growth of 20% compared to the positive control.

Figure 5 summarises the data regarding the BH antimicrobial activity. BH was the most effective against *S. flexneri*, *P. aeruginosa, E. coli, H. influenzae*, and *C. albicans,* with BIR values between 18.65% and 25.42%, with values obtained at 20% concentration tested. There was also a difference between the inhibitory activity on the four mentioned strains. While for *S. flexneri* and *P. aeruginosa,* the inhibitory activity develops alongside concentration, for *E. coli, H. influenzae*, and *C. albicans*, increasing the concentration led to a decrease in the inhibition capacity, proving a strain-boosting effect. Regarding the other tested strains *(S. pyogenes* and *S. aureus*), BH had almost no proven effect, with BIR% ranging from –4.85% to 8.47%. Concerning *C. parapsilopsis*, BH had a proven growth-boosting effect, with an MIR compared to control (%) varying from 0.43% to negative values of –31.32% compared to the positive control. 

*E. coli* and *C. albicans* were the only two ATCC strains on which the inhibitory effect of AH was observed. The evolution of BIR% ranged from 1.42% to 31.13% in the case of *E. coli* with an inhibitory trent alongside the increase in concentration, while for *C. albicans*, the values of BIR ranged from 24.14% to 11.38%. The quantity of honey tested influenced the inhibitory values proving a boosting effect, with MIR decreasing with the increase in concentration (Figure 6).

*S. pyogenes*, *P. aeruginosa*, and *S. typhimurium* proved less sensitive to the effect of AH, starting with negative values for AH 10% and slowly increasing values reaching BIR values ranging from 9.25% to 11.31% for AH 20%.

*S. aureus* and *S. flexneri* demonstrated negative BIR% values, proving a bacterial boosting effect correlated with the increase in concentration. The inhibition rate for *C. parapsilopsis* was concentration-dependent, ranging from −40.93% to −19.98%, proving that *C. parapsilopsis* is sensitive to the AH antimicrobial activity, but the concentrations tested were not sufficient enough to demonstrate positive values. 

Concerning the three LH concentrations tested (LH 10%, LH 15%, and LH 20%), the results presented in Figure 7 show the best antimicrobial effect recorded against *S. flexneri* (with BIR ranging from 8.53% to 17.66%). For *E. coli* and *H. influenzae,* the effect was contrary to the concentration increase, with decreasing BIR values ranging from 26.17% to 5.63%. *S. pyogenes, S. typhimurium*, and *C. albicans* proved less sensitive to LH, with BIR%/MIR% values that showed small inhibitory activity due to insufficient concentration. 

Concerning the inhibition of the clinical *E. coli* isolates, we found a positive correlation of the antimicrobial effect with the concentration, as seen in Figure 8. Of all the honey samples tested on the *E. coli* strains (*E. coli* (BIDLMV code 2023), *E. coli* (BIDLMV code 2067), *E. coli* (BIDLMV code 21117)), MH was the most active, with BIR ranging from 49.2% up to 74.9% in the case of MH 20%. The other types of honey showed similar efficacy values, with the concentration of 10% being too low to determine inhibition in most cases. 

Both *B. cepacia* isolates proved sensitive only at the 20% concentrations tested in the case of all honey types tested, as seen in Figure 9. The values ranged from 7.07% to 16.57% in the case of *B. cepacia* (BIDLMV code 2195) and 9.12% to 17.26% for *B. cepacia* (BIDLMV code 21952). BH did not affect any of the *B. cepacia* isolates tested.

BIR% for *P. vulgaris*, presented in Figure 10, showed that *P. vulgaris* (BIDLMV code 20624) proved resistant to MH at all concentrations tested, with negative values ranging from −32.91% to −25.69%. BH also demonstrated no effect on *P. vulgaris* (BIDLMV code 20624) at 10% and 15%, with positive values only at the 20% concentration tested. AH and LH samples proved effective at 15% and 20%, with inhibitory values ranging from 0.84% to 38.14% for AH 20% and 5.79% to 54.18% for LH 20%.

Concerning *P. aeruginosa* (Figure 11), five clinical isolates were tested. MH demonstrated antibacterial activity on the tested strains, starting with the first concentration tested (10%), unlike the other types of honey that disrupted the increase in the strain mass only from 15%.

The MIC for BH and AH concerning *P. aeruginosa* started at 15%. The inhibition trend is a positive one, the negative effect of the 10% concentration being given only by the insufficient amount tested. This statement is supported by the increasing values of the BIR% at the following concentrations tested. Regarding to the 15%, BH and AH proved inhibitory activity, with values ranging from −4.60% to 30.89% for BH and 1.12% to 20.51% for AH. LH proved to be the less effective against the *P. aeruginosa* clinical resistant isolates, with both 10% and 15% non-active, but with a positive inhibitory trend, the 20% concentration leading to inhibitory rates varying from 12.99% to 60.38%.

To summarise the data presented in Figure 12, all the tested honey samples affected the growth of *M. pachydermatis* (BIDLMV code 20127) and *M. pachydermatis* (BIDLMV code 21521). The difference in efficacy is given by the lack of effect of MH 10%, the effect overwritten by the high inhibitory values achieved by MH 15% and MH 20%. BIR% values of MH were the highest of all honey samples tested for both *M. pachydermatis* strains. All tested honey samples proved positive inhibitory activity, increasing BIR values alongside the increase in concentration.

Two registered effects might occur concerning the effect of the tested honey samples on the bacterial/fungal strains. A strain-boosting effect is where the bacterial/mycelial mass growth of the strain affected by the honey samples is larger than the values obtained in case of the positive control, whereas an inhibitory effect is where the mass growth of the bacterial/mycelial strain is lower than that reported for the positive control, represented by the uninhibited ATCC or clinical isolate taken into analysis.

The yellow colour highlights MIC values (inhibition) but with upward evolution, the effect being one of a sustained strain-boosting effect with increasing concentration; the red colour highlights sustained MIC values, a trend sustained as the concentration increases (Table 1 and Table 2).

Table 1 presents the highlighted OD values corresponding to each ATCC strain tested MIC value.

*S. pyogenes* was inhibited at a MIC value by MH 10%, BH 10% (but with a strain-boosting effect), AH 15%, and LH 20%. 

*S. aureus* was inhibited at a MIC value by MH 10%, BH 10%, and AH 10% (the strain-boosting effect developing values higher than the OD of the positive control for AH 15% and AH 20%). LH had an insufficient inhibitory effect to determine MIC.

*S. flexneri* was inhibited by all honey samples at 10% concentration except for AH. AH had a potentiating effect, determining a strain-boosting effect of the strain.

*P. aeruginosa* was the only ATCC strain that proved resistant to MH at all concentrations tested. Overall, only BH at 10% and AH at 15% proved MIC for *P. aeruginosa*, with MH and LH having no inhibitory effect on the strain.

MH 10% and AH 10% were the MIC for *E. coli,* while BH 10% and LH 10% also inhibited the strain but with a decrease in efficacy alongside the increase in concentration.

For *S. typhimurium*, MH and BH showed MIC at 10%, noting that the effect is a negative inhibitory one, with the inhibition decreasing once the concentration is increased.

*H. influenzae* proved resistant to the honey samples tested, with values of MIC of 10% but with the annotation that all the honey samples tested had a strain-boosting effect which increased with the concentration.

*C. parapsilopsis* was inhibited only by MH 10% and MH 15% (with a strain-boosting trend) and BH 10%, with all other samples showing no antifungal effect.

The antifungal effect was more pronounced in the case of *C. albicans,* with MIC values proved for MH 10%. BH 10% and AH 10% also inhibited mycelial development but with a strain-boosting effect accentuated by increased concentration.

Table 2 presents the OD values recorded for all the resistant clinical isolates in the study. For *E. coli*, all but BH presented the MIC at the 10% concentration, and BH had the MIC starting only at 15%. 

*Burkholderia* strains proved to be the most resistant of all strains tested, with only a small number of MIC. MH demonstrated MIC only at 20%, with the same MIC values appearing for BH, AH, and LH. 

Of all three *Proteus* strains tested, *P. vulgaris* (BIDLMV code 20624) proved to be the most resistant, with only BH developing MIC at 20%, AH at 20%, and LH at 15%. The other two clinical isolates were inhibited starting at 10% in all honey samples tested, except for AH, which proved the MIC at 15%.

Overall, all *Pseudomonas* clinical isolates showed similar sensitivity to the samples tested. In almost all strains tested (except *P. aeruginosa* (BIDLMV code 19117)), MH was the only honey sample with the MIC at 10%, with all the other samples tested inhibiting the strain development only starting at 15%.

In the case of both *M. pachidermitis* isolates, MH developed MIC only starting from 15%; the rest of the honey samples tested proved MIC at the lowest concentration tested (10%).

## 3. Discussion

### 3.1. Antioxidant Profile

The complex biochemical compounds seem to be responsible for the antimicrobial activity of the honey. Although the mechanism of action against bacteria is not yet fully understood, data from the literature suggest an important role of polyphenols in the antimicrobial activity of honey [17,28]. As an exception, the antimicrobial activity of MH is attributed to high concentrations of MGO [18].

The present study shows that MH has the highest content of TPC (528.160 ± 3.043 mg GAE/kg sample) compared to the other samples studied. Data from the literature show variable values of TPC in MH. Alzahrani et al. (2012) [40] reported a value of TPC of 89.90 ± 11.75 mg gallic acid/kg sample, while Bundit et al. (2016) [41] identified a value of 65.79 mg GAE/kg sample, which is close to the present study. The content of total polyphenols of rapeseed honey, linden honey, and acacia honey reported in the literature [42,43,44] are close to the value obtained in this study.

Regarding TFC, the values obtained in the present study are higher than the one reported in the literature [44,45,46]. Yao et al. (2003) [46] reported a flavonoid content in MH of 3.06 mg/100 g honey, while in the present study, the value is double. Similarly, data from the literature reported lower values of TFC in linden honey and rapeseed honey [44,45]. Regarding acacia honey, TFC reported by Dobre et al. (2014) [45] was 13 mg/kg, while a recent study reported a value of 45.714 mg/kg [42] which is almost double compared to the value obtained in the present study.

The values of the antioxidant activity that were observed in the present study are higher than the data reported in the literature. Gośliński et al. (2020) [23] reported a value of antioxidant activity for manuka honey of 40%, although this study showed a value of almost 70% for the same type of honey. For linden honey, the same author found a value of the antioxidant activity of only 14.4% [23], almost four times lower compared to the value obtained in the present study, while Dobre et al. (2014) [45] reported values of 27.2%. The same situation was observed in rapeseed honey and acacia honey. Some authors, evaluating the antioxidant capacity of several types of honey, reported values between 10.7% and 29.98% for acacia honey [45,47], being lower than the value obtained in the present study (47.92%). Due to the factors that influence the components of honey, such as climate and geographical area, the values of the antioxidant capacity of studied honey samples in the western part of Romania are higher than the data from the literature [23,45,47]. 

The present study showed that manuka honey had the highest TPC, TFC, and antioxidant capacity among the four honey samples, followed by linden honey and rapeseed honey. Acacia honey was characterised by the lowest values of TPC, TFC, and antioxidant capacity. Similar results have been reported by other authors, who showed elevated values of TPC and TFC in manuka honey followed by linden honey [23]. Regarding linden honey, rapeseed honey, and acacia honey, data from the literature show similar results [45] to those found in this study, as well as contradictory results. Thus, Kędzierska-Matysek et al. (2021) [44] reported the highest TFC in linden honey and the lowest in acacia honey. However, analysing TPC, the author observed the lowest values in rapeseed honey, not in acacia honey [44]. These findings highlight the diversity of monofloral honey from one region to another.

### 3.2. Antimicrobial Activity

Different types of honey showed significant differences in the antibacterial activity against the tested bacterial strains [23,25,26,27,44]. However, in the literature, there is no solid answer as to what makes certain types of honey better than others [48].

Manuka honey has received particular attention in the West due to its antimicrobial activity. This type of honey is more effective in infections caused by Gram-positive bacteria than Gram-negative ones [24]. Thus, Kumar et al. (2014) [49] demonstrated the inhibitory capacity of manuka honey on all *E. faecalis*, but not for *E. coli* strains, which seem to be more resistant to treatment with this type of honey. Similarly, Grecka et al. (2018) [50] highlighted the fact that the MIC value for *S. aureus* and *S. epidermidis* was 3.125% (*v*/*v*), while for *E. coli* and *P. aeruginosa* the MIC value was higher, reaching 6.25% (*v*/*v*). Another study that demonstrates the higher efficacy of manuka honey against Gram-positive bacteria compared to negative ones is the study conducted by Gkoutzouvelidou et al. (2021) [51], which reports that MIC for *S. aureus* is 25%, while for *S. typhymurium* it is greater than 25%. The present study could not highlight a higher efficacy of manuka honey against Gram-positive strains than negative ones. Except for *B. cepacia*, a MIC of 10% is effective against Gram-positive (*S. aureus*, *S. pyogenes*) and most Gram negative (*E. coli*, *P. aeuginosa*, *P. vulgaris*, *H. influenzae*, *S. flexneri*, and *S. typhymurium*) bacterial strains studied. Similar to our findings, some authors reported a low value of MIC for *S. aureus* and *P. aeruginosa* strains (12.5%) [52], while other studies demonstrated that only an MIC value of 22.5% has an inhibitory effect against both bacterial species [22]. 

The MIC values of manuka honey seem to be variable in the same group of Gram bacteria. If the MIC values for *S. aureus* are between 3.11 and 25% [50,51,52], studies showed that MIC against *S. pyogenes* was at 20% [53]. In contrast, the present study demonstrated a lower MIC value of only 10% for both *S. aureus* and *S. pyogenes*.

The same situation was observed for the group of Gram-negative bacteria. For *P. aeruginosa*, the MIC value seemed to be 10% (*w*/*v*), while 12% (*w*/*v*) had a bactericidal effect against bacteria [54]. The growth and biofilm inhibitory concentration is generally 16%. On the other hand, other studies sustain MIC values against *P. aeruginosa* higher than those mentioned above. More recently, Roberts et al. (2019) [35] claimed that the majority of *P. aeruginosa* isolated strains from cystic fibrosis lung infection were inhibited at a concentration of 32% (*w*/*v*) manuka honey. Mandal et al. (2010) [55] showed that this type of honey had less antimicrobial activity against *P. aeruginosa* compared to *E. coli* strains. The MIC value against *P. aeruginosa* isolates was 3.50% (*v*/*v*), while the value ranged between 3.00% and 3.50% (*v*/*v*) for *E. coli* isolates [55]. These results are similar to the present study, which revealed a MIC of 10% for all *E.coli* strains and 10–15% for all *P. aeruginosa* isolates. Regarding the antimicrobial efficacy against *E. coli* strains, most studies show that the MIC varied between 12.5% and 25% (*v*/*v*) [33,56,57,58], with these values being higher than those obtained in the current study. Similarly, higher MIC values of manuka honey against *Proteus* spp., compared to the results of this study, were reported by other authors. Most of the studies sustain that MIC values of manuka honey vary between 20 and 30% [33,34], with a higher concentration, namely, 50% (*w*/*v*), causing significant partial detachment of *P. mirabilis* biofilm after 24 h [32]. The reduction of viability and virulence of *B. cepacia* strains has been demonstrated at low concentrations of this type of honey [59]. Jenkins et al. (2015) [36] found that the MIC of manuka honey for the 55 isolates of *B. cepacia* was ≤7% (*w*/*v*), with an average of 4.7% (*w*/*v*). These CMI values were lower compared to those obtained for *P. aeruginosa* (MIC of ≤10%, with an average of 7.3%), although *B. cepacia* is more resistant to antibiotics than *P. aeruginosa* [36]. These results contradict those obtained in the present study, demonstrating a higher MIC value of manuka honey (MIC—20%) for *B. cepacia* than *P. aeruginosa* (MIC—10–15%).

The differences between the MIC values of manuka honey, obtained by various authors, can be justified on the one hand by the different methods and techniques chosen to test the strains, and on the other hand by the variety of strains tested. 

Fungal pathogens of the skin, such as *Candida albicans* and dermatophyte species, are less susceptible than bacteria to manuka honey [60]. Fernandes et al. (2021) [52] demonstrated the growth of all *Candida* spp. strains are inhibited by manuka honey and Portuguese honey at high concentrations. Thus, the MIC of MH for *C. albicans* and *C. parasilopsis* was 50% (*w*/*v*) [52]. In the case of *Malassezia* spp. strains, the efficacity of MH has been less studied. Oliveira et al. (2018) [61] demonstrated that all 12 *M. pachydermatis* strains isolated from dogs with otitis externa were susceptible to manuka honey at minimum inhibitory concentrations of 40% (*w*/*v*). In contrast, the results of this study revealed the same lower MIC values for both ATTC strains of *C. parasilopsis* and *C. albicans* of only 10%. The MIC value for *M. pachydermatis* isolates was 15%, lower than the values reported in the literature [61]. 

The antimicrobial activity of other types of honey is debatable and is often compared in the literature to manuka honey. Gośliński et al. (2020) [23] compared the antibacterial efficacy of manuka honey with Polish honey (buckwheat honey, multi-floral honey, linden honey) and found that manuka MGO-400 had an antimicrobial efficacy on both Gram-positive (*S. aureus*, *E. faecalis*, *E. faecium*) and Gram-negative (*E. coli*, *S. typhimurium*, *S. enteridis*, *P. aeruginosa*) tested bacteria at a concentration of 30–40% at the least. The MIC of MH for *S. aureus* was 10–15%, and for *E. coli* and *Salmonella typhymurium* was 20–25%. Unlike this, linden honey inhibited only some of the Gram-positive and Gram-negative strains and only at concentrations of 80–90% [23]. Similarly, the present study demonstrated the antibacterial efficacy of linden honey on Gram-negative isolates from otic secretions, but to a lesser extent for ATTC strains. Referring to ATTC strains, the Gram-positive and some Gram-negative strains, except for *H. influenzae*, *S. flexneri*, and *E.coli*, were resistant to all concentrations of linden honey. Other authors have also highlighted the low efficacy of this type of honey on Gram-positive bacteria. Đogo Mraevi et al. (2020) [62] investigated the biological potential of honey from different regions of Serbia, including acacia honey, linden honey, and rapeseed honey. They discovered that all tested honey exhibited antibacterial activity, with inhibition of bacterial growth generally higher against *E. coli* (80% for linden honey) than against *S. aureus* (less than 50%). In contrast, Grecka et al. (2018) [50] demonstrated that linden honey exhibited high activity, mostly against *Staphylococcus* spp. The MIC values for seven linden kinds of honey (63.6%) were ≤6.25% (*v*/*v*) against all reference staphylococci [50]. Similarly, the efficacy of linden honey is sustained by Farkasovska et al. (2019) [63], who demonstrated that the average MIC values were 7.3% against *S. aureus* and 11.5% against *P. aeruginosa*. Even though the MIC values of linden honey for *S. aureus* remain lower, it seems that MIC values *for E. coli* (MIC—25%) [64], *Haemophilus* spp. (MIC—20.3–40.5%), and *S. pneumoniae* (MIC—21.3%–42.5%) [65] are higher. Unlike the data from the literature, in the present study, concentrations of 10% were active against *H. influenzae* ATTC strain, while for the Gram-negative isolates, MIC values were as follows: for *E. coli*—10%, for *P. vulgaris*—15%, and for both *P. aeruginosa* and *B. cepacia* strains—20%. The growth of the ATTC strains of *S. typhimurium* was not inhibited in the presence of the linden honey, although there are data in the literature that support an MIC for this strain of 0.52% and which instead highlight the resistance of *E coli* strains to this type of honey [48]. 

The studies of the antimicrobial activity of acacia and rapeseed honey exhibited weaker antibacterial activities compared to other types of honey [50,66,67]. The average MIC values for these two types of honey were 18.7–19.1% versus 22.2–22.5% for *S. aureus* and *P. aeruginosa*, respectively [19]. In their study, Masalha et al. (2018) [48] investigated the antioxidant and antimicrobial capacities of various types of honey from different countries. The study reported that acacia honey from Palestine (*Acacia tortilis*) and Bulgari had the same MIC value for both *E. coli* and *Salmonella typhimurium*, 0.52% (*w*/*w*) [48]. However, the present study demonstrated a different MIC for the two bacterial strains, with a higher value for *S. typhimurium* (MIC—20%) than for *E. coli* (MIC—10%). On the other hand, Solayman et al. (2015) [68] evaluated the MIC of acacia honey against Gram-positive (*S. aureus*) and Gram-negative (*E. coli O157:H7* and *S. typhimurium*) bacteria. Their results showed that MIC was 25–50% (*w*/*v*) for *E. coli* O157:H7 and *S. typhimurium*, and 25% (*w*/*v*) for *S. aureus* [68]. The present study shows that acacia honey has antimicrobial activity against these bacteria at different concentrations, higher for *S. typhimurium* (MIC—20%) and lower for *E. coli* (MIC—10%). *S. aureus* ATTC strains were inhibited at 10% concentration of acacia honey, and over this value, the boosting effect was observed.

Data from the literature on the antifungal activity of linden honey, acacia honey, and rapeseed honey are limited. These were not tested on *Malassezia* spp. strains, but only on *Candida* spp. strains. Aurongzeb et al. (2019) [69] showed the antifungal activity of acacia honey at minimum inhibitory concentration values of 3–10% *w*/*v* against clinically isolated specimens of *Candida* spp., although other authors reported that no honey tested was effective for *C. albicans* [48]. In the present study, from the three types of local honey, only AH and BH inhibited the growth of *Candida albicans* with MIC of 10%, and over this value, the boosting effect appeared. None of these honeys showed anti-mycelial activity against *C. parapsilosus*. However, the local honey inhibited the growth of *M. pachydermatis* at a concentration of only 10%.

A large study from Romania involved the characterisation of the antimicrobial capacity of 10 types of honey from Transylvania. All the honey samples were tested against certain reference microorganisms, such as *E. coli*, *S. aureus*, *B. cereus*, *P. aeruginosa*, and *Candida albicans*. Compared with ciprofloxacin, all the honey samples showed lower antimicrobial activity than the chosen antibiotic, except for *C. albicans*. The *E. coli* strain showed resistance to the acacia honey samples and one of the linden honeys, all collected from Cluj. The same resistance was noticed in the case of the *C. albicans* strain. The inhibitory effect of honey at 1/1 and 1/4 dilutions was observed in all samples against all bacterial strains. The MIC values of acacia and linden honey was 25% (*w*/*v*). The 1/16 and 1/32 dilutions produced partial effects only in certain bacterial strains, while the 1/64 dilution had no antimicrobial effect on any strain [70].

In contrast, the present study showed that acacia honey and linden honey from the western part of Romania had antimicrobial activity against *E. coli* strains with an MIC value of 10%. Rapeseed honey is active against the same isolates at a concentration of 15%. Although linden honey does not have anti-mycelial activity against the *Candida albicans* ATTC strains, the other two local honeys inhibited the growth of this strain at a concentration of 10%. These different results emphasise the diversity of monofloral honey depending on the geographical area and the climatic conditions for samples collected from the same country.

Correlations between variables:

The analysis of correlation (Table 3) highlights a strong (r > −0.7) negative correlation between
-*S. pyogenes* and TPC (r = −0.776), DPPH (r = −0.733);-*H. influenzae* and TPC (r = −0.890), TFC (r = −0.884) and DPPH (r = −0.889);-*C. parapsilopsis* and TPC (r = −0.808), TFC (r = −0.713) and DPPH (r = −0.780);

A moderate (r > −0.5) negative correlation was registered between the pairs:-*S. aureus* and TPC (r = −0.686) and DPPH (r = −0.620);-*S. flexneri* and TFC (r = −0.539).

A moderate positive correlation was registered between
-*P. aeruginosa* and TFC (r = 0.532).

Mutual correlations between individual variables were determined, proving the dependence between the composition of antimicrobial activity and antioxidant activity of the analysed honey sampler.

Table 4 highlighted a moderate (r > −0.5) negative correlation between *E. coli* (BIDLMV 21117) and TPC (r = −0.517), TFC (r = −0.507), and DPPH (r = −0.531).

## 4. Materials and Methods

The study was performed on 4 types of honey: one commercial honey, manuka honey (MH) with MGO 500 (Auribee, Braguta’s Queens L.T.D, Tauranga, New Zealand), and the last three types of honey were represented by acacia honey (AH), linden honey (LH), and rapeseed honey (BH), all collected from a beekeeper in the western part of Romania (LAT: 45.70963, LONG: 21.03042).

### 4.1. Antioxidant Profile

#### 4.1.1. Preparation of the Alcoholic Extract

The preparation of the alcoholic extract was the first step for the antioxidant profile characterisation of honey. A total of 1 g of honey was extracted from each sample with 10 mL 70% ethanol (Sigma-Aldrich; Merck KGaA, Darmstadt, Germany) for 30 min at room temperature using an ultrasonic water bath (FALC Instruments, Treviglio, Italy). After stirring, the extracts were then filtered using Whatman membrane filters, nylon 0.45 µm with 30 mm diameter (Sigma-Aldrich; Merck KGaA, Darmstadt, Germany), and stored at 2–4 °C for subsequent chemical and antimicrobial analyses.

#### 4.1.2. Determination of Total Polyphenol Content by Folin–Ciocâlteu Assay

The total phenol content was determined by the Folin–Ciocâlteu method with minor modifications [71]. Thus, a quantity of 0.5 mL was taken from each extract prepared as described above, over which 1.25 mL of Folin–Ciocâlteu reagent (Sigma-Aldrich Chemie GmbH, Munich, Germany) was added, diluted 1:10 with distilled water. The samples were incubated for 5 min at room temperature. After incubation, 1 mL of Na_2_CO_3_ (Geyer GmbH, Renningen, Germany; 60 g/L aqueous solution) was added. The samples were incubated for 30 min at 50 °C, using the thermostat (INB500, Memmert GmbH, Schwabach, Germany). The absorbance was read at 750 nm using a UV–VIS spectrophotometer (Specord 205; Analytik Jena AG, Jena, Germany). As a control, we used 70% ethanol (Sigma-Aldrich; Merck KGaA, Darmstadt, Germany). The calibration curve was obtained with gallic acid (2.5–250 μg/mL). Results were expressed in mg GAE per kg dry matter (d.m.). All determinations were made in triplicate.

#### 4.1.3. Determination of Total Flavonoid Content (TFC)

Total flavonoid content in honey was determined according to the method described by Al-Farsiet et al. (2018) [72], with smallionsion modification. From each sample, 1.5 mL of extract was taken, adding 4.5 mL of distilled water, and then mixed with 1 mL NaNO_2_ 5% solution. After 6 min of incubation, 1 mL of 10% Al(NO_3_)_3_ was added to the mixture. After another 6 min of incubation, we added 10 mL of 4% NaOH solution. The volume was then increased to 25 mL by the addition of 60% of ethanol solution. After 15 min, the absorbance of the mixture solution was measured with a UV–VIS spectrophotometer (Specord 205; Analytik Jena AG, Jena, Germania) at 510 nm against a control containing ethanol 70%. The results were expressed as mg QE/100 g, and all determinations were performed in triplicate. The calibration curve was obtained with quercetin (concentration range: 0.5–50 μg/mL).

#### 4.1.4. Antioxidant Capacity by 1,1-Diphenyl-2-picrylhydrazyl (DPPH) Assay

The antioxidant capacity of the honey samples was determined according to Chen’s method [73] with minor modifications, using a 0.03 mM ethanolic solution of 1,1-diphenyl-2-picrylhydrazyl (DPPH, Sigma-Aldrich, Taufkirchen, Germany). It is widely used to test the antioxidant capacity of foods, especially fruits, vegetables, juices, and even honey [74,75,76,77]. A total of 1 mL of each prepared honey extract was taken, to which 2.5 mL of DPPH solution (Sigma-Aldrich, Taufkirchen, Germany) was added. The samples were shaken and then incubated in the dark for 30 min at room temperature. The absorbance was measured with a UV–VIS spectrophotometer (Specord 205; Analytik Jena AG, Jena, Germany) at 518 nm. An average value was calculated from the three measurements performed for each honey sample. The control was prepared under the same conditions, the difference being that, instead of the honey solution, the same volume of distilled water was used. The antioxidant activity of each sample was calculated as a percentage of the CSR (the p overof radical uptake capacity). CSR of the tested samples was calculated according to the following equation: (1)$$\mathrm{CSR}\;(\%)=\frac{A_{control}-A_{sample}}{A_{control}}\times 100$$
where
**A_control_**—the control absorbance values;**A_sample_**—the absorbance values tested of the samples.

### 4.2. Antimicrobial Activity

To test the antimicrobial activity, aqueous extracts of each honey sample were prepared by mixing 1 g of honey with 1 mL of sterile distilled water, and then different quantities were spotted into the 96 well plates in order to reach the 10%, 15%, and 20% concentrations selected by our study. 

The microbial reference strains (ATCC) used in this study were obtained from the culture collection of the Microbiology Laboratory of the Interdisciplinary Research Platform within the University of Agricultural Sciences and Veterinary Medicine “King Mihai I of Romania” in Banat, Timisoara. 

The samples were also tested against clinically, drug-resistant strains, isolated from recurrent dog ear infections, which are part of the culture collection of the Bacterial Infectious Diseases Laboratory belonging to the Infectious Diseases Clinic within the Faculty of Veterinary Medicine, University of Agricultural Sciences and Veterinary Medicine “King Mihai I of Romania” in Banat, Timisoara. In the laboratory, the strains are maintained at −50 °C. After the laboratory acronym, isolates from otic secretions of dogs are abbreviated with BIDLMV, followed by the number of the strain. 

The honey samples were tested against the following reference strains: *Staphylococcus aureus* (ATCC 25923), *Streptococcus pyogenes* (ATCC 19615), *Escherichia coli* (ATCC 25922), *Pseudomonas aeruginosa* (ATCC 27853), *Shigella flexneri* (ATCC 12022), *Salmonella typhimurium* (ATCC 14028), *Haemophilus influenzae* type B (ATCC 10211), *Candida albicans* (ATCC 10231), *Candida parapsilopsis* (ATCC 22019), *E.coli* (BIDLMV code 2023), *E. coli* (BIDLMV code 2067), *E. coli* (BIDLMV code 21117), *Malassezia pachydermatis* (BIDLMV code 20127), *M. pachydermatis* (BIDLMV code 21521), *Burkholderia cepacia* (BIDLMV code 2195), *B. cepacia* (BIDLMV code 21952), *Proteus vulgaris* (BIDLMV code 20624), *P. vulgaris* (BIDLMV code 20124), *P. vulgaris* (BIDLMV code 2147), *P. aeruginosa* (BIDLMV code 19117), *P. aeruginosa* (BIDLMV code 191211), *P. aeruginosa* (BIDLMV code 20129), *P. aeruginosa* (BIDLMV code 20722), and *P. aeruginosa* (BIDLMV code 21615).

The MIC is defined as the lowest compound concentration that yields no visible microorganism growth. The method of MIC determination based on the microbial mass loss by measurement of OD by spectrophotometry according to ISO 20776–1:2019 was described in our previous research [71,75,78].

#### 4.2.1. Bacterial Culture

A 10^−3^ dilution of the fresh culture was used to perform the assay, an inoculum equivalent to a 0.5 McFarland standard. The bacterial strains were revived by overnight growth in brain heart infusion (BHI) broth (Oxoid, CM1135) at 37 °C and, subsequently, passed on BHI Agar (Oxoid, CM1136) for 24 h at 37 °C. The cultures were then diluted at an optical density (OD) of 0.5 McFarland standard (1.5  ×  10^8^ UFC × mL) using BHI broth and a McFarland Densitometer (Grand-Bio, England). The suspensions were tested by spotting 100 μL of microbial suspension in each well of the 96 microdilution well plate, using a Calibra digital 852 multichannel pipette. The tested honey samples were added into wells at concentrations of 10%, 15%, and 20%. The plates were covered and left 24 h at 37 °C. After 24 h, the OD was measured at 540 nm using an ELISA reader (BIORAD PR 1100, Hercules, CA, USA). Triplicate tests were performed for all samples. The suspensions of strain and BHI were used as a negative control. 

To interpret the results, two indicators were calculated BGR and BIR by using the formulas (2) and (3):(2)$$\mathrm{BGR}\;=\frac{{\mathrm{OD}}_{\mathrm{sample}}}{{\mathrm{OD}}_{\mathrm{negativecontrol}}}\times 100\;\left(\%\right)$$
BIR = 100 − BGR (%) (3)
where
OD_sample_—optical density at 540 nm as the mean value of triplicate readings for EOs in the presence of the selected bacteria;OD_negative control_—optical density at 540 nm as the mean value of triplicate readings for the selected bacteria in BHI.

#### 4.2.2. Fungal Culture

A 10^−2^ dilution of the fresh culture was used to perform the assay, an inoculum equivalent to a 0.5 McFarland standard. The ATCC fungal strains were revived by overnight growth in brain heart infusion (BHI) broth (Oxoid, CM1135) at 37 °C and, subsequently, passed on BHI Agar (Oxoid, CM1136) for 48 h at 37 °C. The cultures were then diluted at an OD of 0.5 McFarland standard using BHI broth, a value determined by using a McFarland Densitometer (Grand-Bio, England). The honey samples were tested by placing 100 μL of fungal suspension into each well of the 96-microdilution-well plate. The EOs were tested at concentrations of 2%, 4%, 8%, and 16%, added in each well. The plates were covered and left for 48 h at 37 °C. After 48 h, the OD was measured at 540 nm. Triplicate tests were performed for all samples.

To interpret the results, two indicators were calculated, MGR and MIR, using the following formulas (Formulas (4) and (5)):(4)$$\mathrm{MGR}\;=\frac{{\mathrm{OD}}_{\mathrm{sample}}}{{\mathrm{OD}}_{\mathrm{negativecontrol}}}\times 100\;\left(\%\right)$$
MIR = 100 − MGR (%) (5)
where
OD_sample_—optical density at 540 nm as the mean value of triplicate readings for EOs in the presence of the selected fungi;OD_negative control_—optical density at 540 nm as the mean value of triplicate readings for the selected fungi in BHI.

#### 4.2.3. Statistical Analysis

All determinations were made in triplicate, and the results are reported as mean values ± standard deviation (SD). 

For total polyphenol content, the mean values and standard deviations of all replicates were calculated using GraphPad Prism (v.5.0 software, Manufacture, San Diego, CA, USA). Antimicrobial activity rates, chemical data, figures, and statistical correlation were performed using Microsoft Excel 365.

## 5. Conclusions

The antioxidant potential characterisation revealed that MH contained the highest antioxidant capacity (DPPH: 68,247 ± 0.06%) and highest TPC (528,160 ± 3043 mg GAE/kg sample) and TFC (65,563 ± 0.664 mg/kg sample), followed by LH and BH. AH was proven to have the lowest values of antioxidant capacity, TPC, and TFC.

Even characterised by the highest values of TPC and TFC, MH was proven to be effective on *E. coli* and *B. cepacia* clinical isolates at the same concentration as LH and AH, with further studies being necessary to decipher the synergistic/antagonistic reactions that appeared. The MIC of MH against all *P.aeruginosa* isolates was similar to AH at 15%. All the isolates of *P. vulgaris* were inhibited by 15% LH and 20% AH, with *P. vulgaris* (BIDLMV code 20624) being the most resistant strain. Regarding *M. pachydermatis* isolates, the local types of honey were active at a concentration of 10%, while MH was at a concentration of 15%. Against the ATCC strains, MH was the most effective at MIC of 10%, inhibiting all the strains tested, except for *P. aeruginosa.* BH was also active against ATTC strains at the same concentration as MH, emphasising that the boosting effect may occur above this concentration for some strains.

LH and AH showed antimicrobial activity against all Gram-negative clinical isolates at different concentrations depending on the strain responsible for the infection. However, the analysis of the antioxidant profile showed that LH had the highest content of TPC and TFC after MH, while AH had the lowest values for these biochemical compounds. This fact draws attention to the reconsideration of factors involved in the antimicrobial activity of honey so that in future, more studies would be needed to understand the mechanism by which this natural product inhibits microbial growth.

As a general conclusion, the present study revealed the effectiveness of MH and local honey against ear isolates, suggesting that honey may be considered an alternative or complementary antimicrobial strategy to the allopathic treatment of recurrent dog ear infections.

## Figures and Tables

**Figure 1 antibiotics-11-00595-f001:**
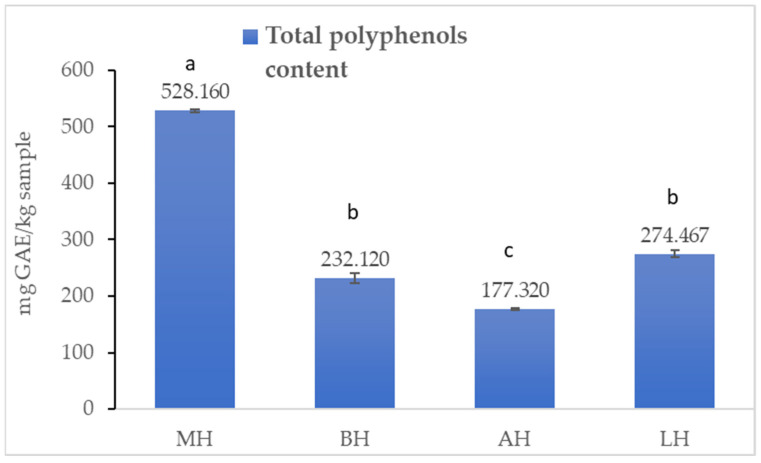
Total polyphenolic content of the honey samples. Mean values are expressed as mg gallic acid equivalent GAE/kg sample. The error bars indicate the standard deviation. Different letters among samples indicate significant differences (*p* < 0.05) among values according to the *t*-test.

**Figure 2 antibiotics-11-00595-f002:**
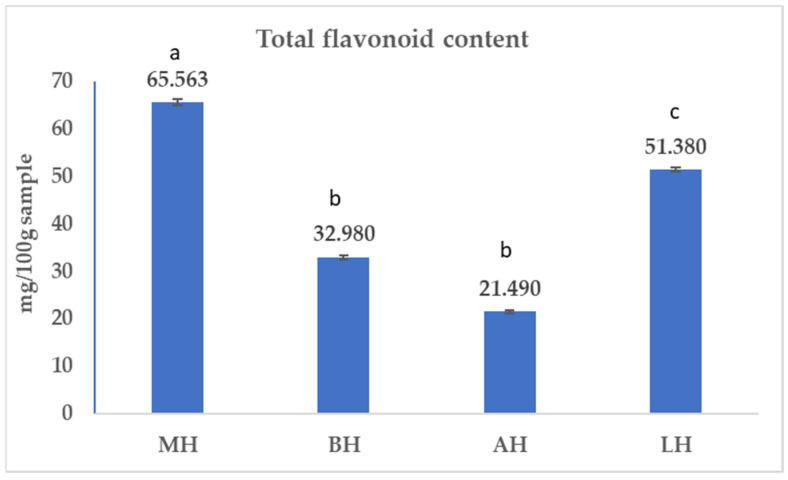
Total flavonoid content (TFC) of the honey sample. Mean values are expressed as mg/kg sample. The error bars indicate the standard deviation. Different letters among samples indicate significant differences (*p* < 0.05) among values according to the *t*-test.

**Figure 3 antibiotics-11-00595-f003:**
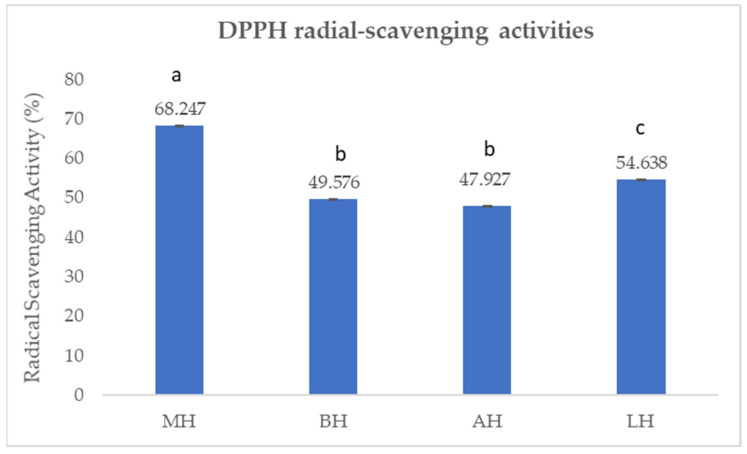
DPPH radical scavenging of the honey samples. Mean values expressed as %. The error bars indicate the standard deviation. Different letters among samples indicate significant differences (*p* < 0.05) among values according to the t-test.

**Figure 4 antibiotics-11-00595-f004:**
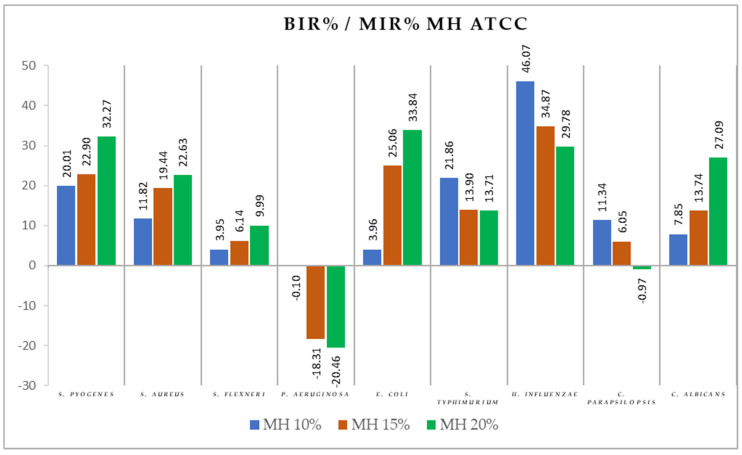
MH antimicrobial activity (expressed as BIR%/MIR%) on ATCC strains.

**Figure 5 antibiotics-11-00595-f005:**
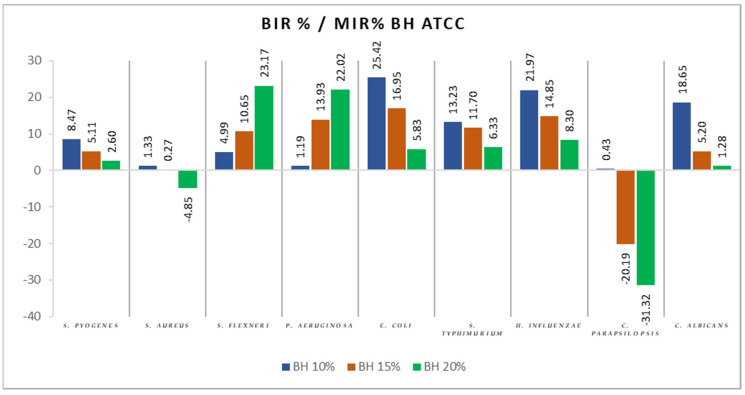
BH antimicrobial activity (expressed as BIR%/MIR%) on ATCC strains.

**Figure 6 antibiotics-11-00595-f006:**
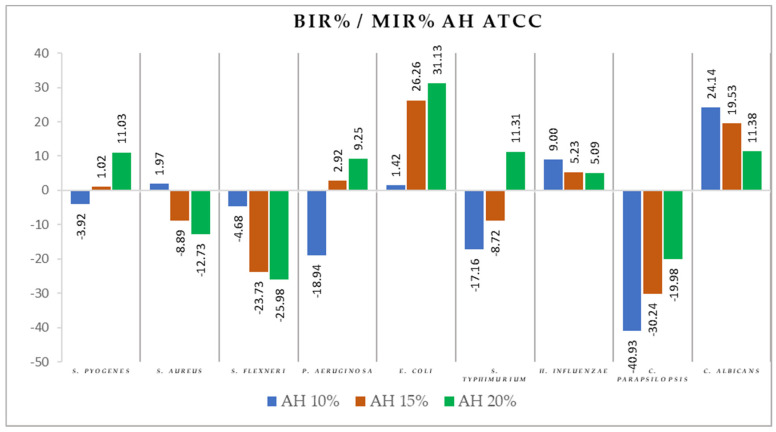
AH antimicrobial activity (expressed as BIR%/MIR%) on ATCC strains.

**Figure 7 antibiotics-11-00595-f007:**
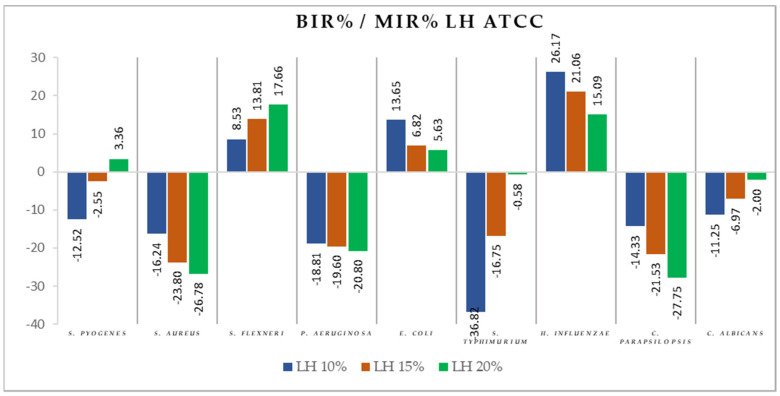
LH antimicrobial activity (expressed as BIR%/MIR%) on ATCC strains.

**Figure 8 antibiotics-11-00595-f008:**
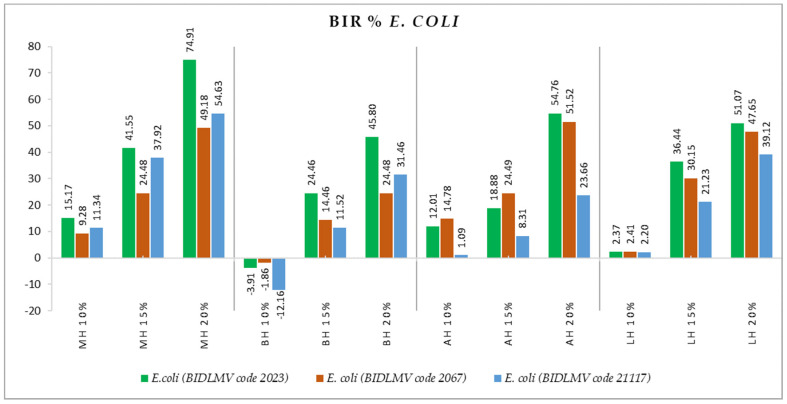
Antimicrobial activity (expressed as BIR%) of honey samples tested on *E. coli* isolates.

**Figure 9 antibiotics-11-00595-f009:**
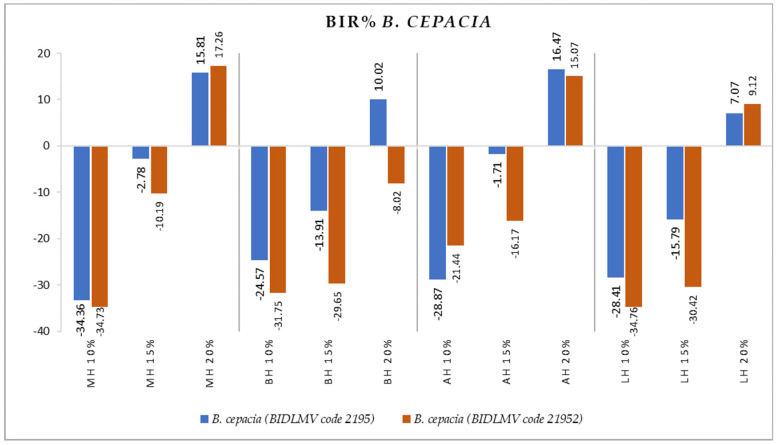
Antimicrobial activity (expressed as BIR%) of honey samples tested on *B. cepacia* isolates.

**Figure 10 antibiotics-11-00595-f010:**
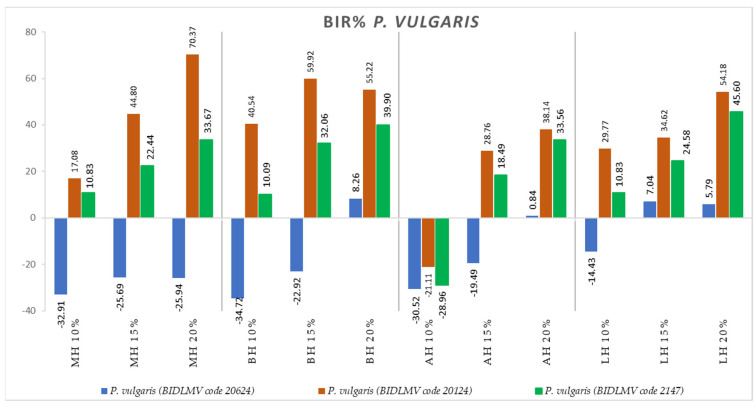
Antimicrobial activity (expressed as BIR%) of honey samples tested on *P. vulgaris* isolates.

**Figure 11 antibiotics-11-00595-f011:**
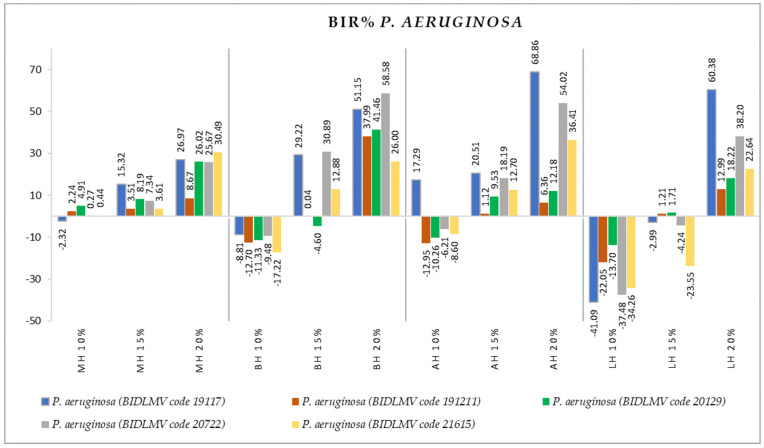
Antimicrobial activity (expressed as BIR%) of honey samples tested on *P. aeruginosa* isolates.

**Figure 12 antibiotics-11-00595-f012:**
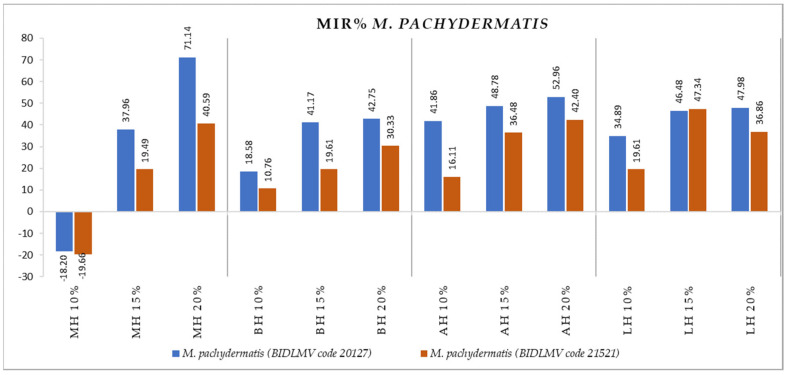
Antimicrobial activity (expressed as MIR%) of honey samples tested on *P. aeruginosa* isolates.

**Table 1 antibiotics-11-00595-t001:** The OD reading of honey samples tested on the ATTC strains.

The Concentration of Honey %	*S. pyogenes* (ATCC 19615)	*S. aureus* (ATCC 25923)	*S. flexneri* (ATCC 12022)	*P. aeruginosa* (ATCC 27853)	*E. coli* (ATCC 25922)	*S. typhimurium* (ATCC 14028)	*H. influenzae* (ATCC 10211)	*C. parapsilopsis* (ATCC 22019)	*C. albicans* (ATCC 10231)
MH 10	**0.626 ± 0.006**	**0.552 ± 0.002**	**0.923 ± 0.002**	1.006 ± 0.002	**1.262 ± 0.002**	**0.815 ± 0.004**	**0.773 ± 0.008**	**0.821 ± 0.007**	**0.939 ± 0.003**
MH 15	0.604 ± 0.004	0.504 ± 0.003	0.902 ± 0.002	1.189 ± 0.002	0.985 ± 0.004	0.898 ± 0.004	0.934 ± 0.005	0.870 ± 0.007	0.879 ± 0.003
MH 20	0.530 ± 0.004	0.484 ± 0.002	0.865 ± 0.002	1.211 ± 0.004	0.869 ± 0.003	0.900 ± 0.004	1.007 ± 0.005	0.935 ± 0.007	0.743 ± 0.003
BH 10	**0.717 ± 0.004**	**0.618 ± 0.005**	**0.913 ± 0.002**	**0.993 ± 0.002**	**0.980 ± 0.003**	**0.905 ± 0.004**	**1.119 ± 0.005**	**0.922 ± 0.005**	**0.829 ± 0.003**
BH 15	0.743 ± 0.004	0.624 ± 0.003	0.859 ± 0.007	0.865 ± 0.003	1.091 ± 0.003	0.921 ± 0.004	1.221 ± 0.005	1.113 ± 0.005	0.966 ± 0.003
BH 20	0.763 ± 0.005	0.656 ± 0.008	0.738 ± 0.003	0.784 ± 0.005	1.237 ± 0.005	0.977 ± 0.004	1.315 ± 0.005	1.216 ± 0.005	1.006 ± 0.003
AH 10	0.814 ± 0.003	**0.614 ± 0.003**	1.006 ± 0.002	1.195 ± 0.003	**1.295 ± 0.003**	1.222 ± 0.004	**1.305 ± 0.005**	1.305 ± 0.005	**0.773 ± 0.003**
AH 15	**0.775 ± 0.007**	0.682 ± 0.004	1.189 ± 0.002	**0.976 ± 0.002**	0.969 ± 0.005	1.134 ± 0.004	1.359 ± 0.005	1.206 ± 0.005	0.820 ± 0.003
AH 20	0.697 ± 0.005	0.706 ± 0.006	1.211 ± 0.004	0.912 ± 0.003	0.905 ± 0.003	**0.925 ± 0.003**	1.361 ± 0.005	1.111 ± 0.005	0.903 ± 0.003
LH 10	0.881 ± 0.006	0.728 ± 0.005	**0.879 ± 0.004**	1.194 ± 0.005	**1.135 ± 0.004**	1.427 ± 0.004	**1.059 ± 0.005**	1.059 ± 0.005	1.134 ± 0.003
LH 15	0.803 ± 0.004	0.775 ± 0.004	0.828 ± 0.003	1.202 ± 0.005	1.224 ± 0.003	1.218 ± 0.014	1.132 ± 0.005	1.125 ± 0.016	1.090 ± 0.077
LH 20	**0.757 ± 0.004**	0.794 ± 0.006	0.791 ± 0.003	1.214 ± 0.003	1.240 ± 0.003	**1.049 ± 0.004**	1.218 ± 0.003	1.183 ± 0.032	1.039 ± 0.005
C	0.783 ± 0.004	0.626 ± 0.005	0.961 ± 0.005	1.005 ± 0.005	1.314 ± 0.004	1.043 ± 0.004	1.434 ± 0.005	0.926 ± 0.005	1.019 ± 0.004

The red colour highlights the samples where the MIC was determined, with the MIC value highlighted. The effect was maintained together with an increase in concentration. The samples with a strain-boosting effect maintained with increased concentration but that still reached a MIC value are marked in yellow.

**Table 2 antibiotics-11-00595-t002:** The OD values of honey samples tested on the isolated strains.

Honey Concentration%	*E. coli* (BIDLMV 2023)	*E. coli* (BIDLMV 2067)	*E. coli* (BIDLMV 21117)	*B. cepacia* (BIDLMV 219)	*B. cepacia* (BIDLMV 21952)	*P. vulgaris* (BID-LMV 20624)	*P. vulgaris* (BIDLMV 20124)	*P. vulgaris* (BIDLMV 2147)	*P. aeruginosa* (BIDLMV 19117)	*P. aeruginosa* (BIDLMV 19121)	*P. aeruginosa* (BIDLMV 20129)	*P. aeruginosa* (BIDLMV 20722)	*P. aeruginosa* (BIDLMV 21615)	*M. pachydermatis* (BIDLMV 20127)	*M. pachydermatis* (BIDLMV 21521)
MH 10	**1.577 ± 0.005**	**1.721 ± 0.004**	**1.436 ± 0.007**	2.045 ± 0.004	1.925 ± 0.006	1.889 ± 0.008	**1.270 ± 0.003**	**1.326 ± 0.005**	1.367 ± 0.005	**1.568 ± 0.002**	**1.634 ± 0.004**	**1.332 ± 0.003**	**1.285 ± 0.005**	1.771 ± 0.007	1.672 ± 0.004
MH 15	1.087 ± 0.006	1.433 ± 0.006	1.006 ± 0.009	1.564 ± 0.006	1.575 ± 0.007	1.786 ± 0.005	0.846 ± 0.004	1.153 ± 0.004	**1.131 ± 0.007**	1.548 ± 0.008	1.577 ± 0.003	1.238 ± 0.003	1.244 ± 0.004	**0.929 ± 0.002**	**1.125 ± 0.006**
MH 20	0.466 ± 0.004	0.964 ± 0.004	0.735 ± 0.008	**1.281 ± 0.006**	**1.182 ± 0.006**	1.790 ± 0.004	0.454 ± 0.003	0.986 ± 0.004	0.976 ± 0.003	1.465 ± 0.005	1.271 ± 0.003	0.993 ± 0.005	0.897 ± 0.005	0.432 ± 0.003	0.830 ± 0.003
BH 10	1.932 ± 0.005	1.932 ± 0.007	1.817 ± 0.053	1.896 ± 0.005	1.883 ± 0.004	1.914 ± 0.005	**0.911 ± 0.003**	**1.337 ± 0.005**	1.454 ± 0.005	1.808 ± 0.004	1.913 ± 0.005	1.463 ± 0.005	1.513 ± 0.006	**1.220 ± 0.005**	**1.247 ± 0.040**
BH 15	**1.404 ± 0.004**	**1.623 ± 0.009**	**1.433 ± 0.006**	1.734 ± 0.003	1.853 ± 0.005	1.747 ± 0.005	0.614 ± 0.005	1.010 ± 0.004	**0.946 ± 0.006**	**1.603 ± 0.004**	1.797 ± 0.007	**0.923 ± 0.004**	**1.125 ± 0.005**	0.881 ± 0.003	1.123 ± 0.006
BH 20	1.008 ± 0.007	1.433 ± 0.005	1.110 ± 0.005	**1.370 ± 0.005**	**1.544 ± 0.005**	**1.304 ± 0.006**	0.686 ± 0.005	0.894 ± 0.003	0.653 ± 0.003	0.995 ± 0.004	**1.006 ± 0.004**	0.553 ± 0.003	0.955 ± 0.005	0.858 ± 0.003	0.973 ± 0.007
AH 10	**1.636 ± 0.006**	**1.617 ± 0.006**	**1.602 ± 0.009**	1.961 ± 0.008	1.735 ± 0.006	1.855 ± 0.006	1.855 ± 0.004	1.918 ± 0.006	**1.105 ± 0.009**	1.812 ± 0.003	1.894 ± 0.005	1.419 ± 0.004	1.402 ± 0.004	**0.871 ± 0.007**	**1.172 ± 0.007**
AH 15	1.508 ± 0.006	1.432 ± 0.006	1.485 ± 0.007	1.548 ± 0.008	1.660 ± 0.006	1.698 ± 0.005	**1.091 ± 0.006**	**1.212 ± 0.006**	1.062 ± 0.004	**1.586 ± 0.003**	**1.554 ± 0.005**	**1.093 ± 0.005**	**1.127 ± 0.004**	0.767 ± 0.004	0.887 ± 0.006
AH 20	0.841 ± 0.009	0.920 ± 0.008	1.237 ± 0.006	**1.271 ± 0.007**	**1.214 ± 0.005**	**1.409 ± 0.006**	0.948 ± 0.003	0.988 ± 0.005	0.416 ± 0.005	1.502 ± 0.012	1.509 ± 0.008	0.614 ± 0.004	0.821 ± 0.004	0.705 ± 0.003	0.805 ± 0.008
LH 10	**1.815 ± 0.006**	**1.851 ± 0.007**	**1.584 ± 0.018**	1.954 ± 0.008	1.926 ± 0.007	1.626 ± 0.005	**1.076 ± 0.006**	**1.326 ± 0.002**	1.885 ± 0.004	1.958 ± 0.004	1.953 ± 0.009	1.837 ± 0.003	1.733 ± 0.008	**0.975 ± 0.004**	**1.123 ± 0.004**
LH 15	1.182 ± 0.007	1.325 ± 0.005	1.276 ± 0.007	1.762 ± 0.009	1.864 ± 0.004	**1.321 ± 0.003**	1.002 ± 0.005	1.122 ± 0.006	1.376 ± 0.005	**1.585 ± 0.005**	**1.689 ± 0.003**	1.393 ± 0.006	1.595 ± 0.006	0.802 ± 0.007	0.736 ± 0.345
LH 20	0.910 ± 0.008	0.993 ± 0.006	0.986 ± 0.007	**1.414 ± 0.008**	**1.299 ± 0.004**	1.339 ± 0.003	0.702 ± 0.008	0.809 ± 0.008	**0.529 ± 0.003**	1.396 ± 0.005	1.405 ± 0.004	**0.826 ± 0.005**	**0.999 ± 0.006**	0.779 ± 0.005	0.882 ± 0.007
C	1.859 ± 0.005	1.897 ± 0.005	1.620 ± 0.008	1.522 ± 0.003	1.429 ± 0.002	1.427 ± 0.005	1.532 ± 0.005	1.487 ± 0.005	1.337 ± 0.002	1.605 ± 0.006	1.719 ± 0.003	1.336 ± 0.002	1.292 ± 0.002	1.499 ± 0.004	1.397 ± 0.005

The red colour highlights the samples where the MIC was determined, with the MIC value highlighted. The effect was maintained together with an increase in concentration. The samples with a strain-boosting effect maintained with increased concentration but that still reached a MIC value are marked in yellow.

**Table 3 antibiotics-11-00595-t003:** Pearson correlation coefficient matrix for ATCC antimicrobial and antioxidant activity of honey samples.

	Pyogenes	Staphilococcus	Shigella	*Pseudomonas*	*E. coli*	Salmonella	*H. inflenzae*	Parapsilopsis	*C. albicans*	TPC	TFC	DPPH
Pyogenes	1											
Staphilococcus	0.774	1.000										
Shigella	−0.001	0.006	1.000									
*Pseudomonas*	−0.025	0.009	−0.163	1.000								
*E coli*	0.513	0.364	−0.470	0.090	1.000							
Salmonella	0.823	0.588	0.014	0.430	0.339	1.000						
*H. influenzae*	0.523	0.531	0.357	−0.377	−0.038	0.292	1.000					
Parapsilopsis	0.696	0.608	0.130	−0.127	0.366	0.546	0.876	1.000				
*C albicans*	0.556	0.681	−0.462	0.011	0.497	0.434	−0.058	0.128	1.000			
TPC	−0.776	−0.686	−0.321	0.367	−0.145	−0.457	−0.890	−0.808	−0.150	1.000		
TFC	−0.522	−0.361	−0.539	0.532	0.036	−0.197	−0.884	−0.713	0.207	0.908	1.000	
DPPH	−0.733	−0.620	−0.305	0.448	−0.119	−0.380	−0.889	−0.780	−0.103	0.993	0.931	1.000

Red colored values highlight the existence of significant correlations between variables.

**Table 4 antibiotics-11-00595-t004:** Pearson correlation coefficient matrix for antimicrobial activity on the clinical isolates and antioxidant activity of honey samples.

	*E. coli* (BIDLMV 2023)	*E. coli* (BIDLMV 2067)	*E. coli* (BIDLMV 21117)	*B. cepacia* (BIDLMV 219)	*B. cepacian* (BIDLMV 21952)	*P. vulgaris* (BID-LMV 20624)	*P. vulgaris* (BIDLMV 20124)	*P. vulgaris* (BIDLMV 2147)	*P. aeruginosa* (BIDLMV 19117)	*P. aeruginosa* (BIDLMV 19121)	*P. aeruginosa* (BIDLMV 20129)	*P. aeruginosa* (BIDLMV 20722)	*P. aeruginosa* (BIDLMV 21615)	*M. pachydermatis* (BIDLMV 20127)	*M. pachydermatis* (BIDLMV 21521)	TPC	TFC	DPPH
*E. coli* (BIDLMV 2023)	**1**																	
*E. coli* (BIDLMV 2067)	**0.932**	**1**																
*E. coli* (BIDLMV 21117)	**0.949**	**0.826**	**1**															
*E. coli* (BIDLMV 21117)	**0.689**	**0.723**	**0.578**	**1**														
*B. cepacia* (BIDLMV 21952)	**0.661**	**0.750**	**0.528**	**0.919**	**1**													
*P. vulgaris* (BID-LMV 20624)	**0.761**	**0.746**	**0.700**	**0.575**	**0.619**	**1**												
*P. vulgaris* (BIDLMV 20124)	**0.718**	**0.748**	**0.666**	**0.475**	**0.461**	**0.907**	**1**											
*P. vulgaris* (BIDLMV 2147)	**0.337**	**0.400**	**0.296**	**0.240**	**0.511**	**0.542**	**0.401**	**1**										
*P. aeruginosa* (BIDLMV 19117)	**0.649**	**0.504**	**0.645**	**0.502**	**0.491**	**0.529**	**0.304**	**0.167**	**1**									
*P. aeruginosa* (BIDLMV 19121)	**0.703**	**0.634**	**0.683**	**0.426**	**0.534**	**0.658**	**0.428**	**0.487**	**0.903**	**1**								
*P. aeruginosa* (BIDLMV 20129)	**0.707**	**0.763**	**0.569**	**0.474**	**0.476**	**0.750**	**0.715**	**0.409**	**0.372**	**0.553**	**1**							
*P. aeruginosa* (BIDLMV 20722)	**0.652**	**0.521**	**0.649**	**0.224**	**0.301**	**0.691**	**0.512**	**0.538**	**0.485**	**0.662**	**0.718**	**1**						
*P. aeruginosa* (BIDLMV 21615)	**0.770**	**0.646**	**0.775**	**0.372**	**0.405**	**0.773**	**0.641**	**0.460**	**0.541**	**0.674**	**0.684**	**0.943**	**1**					
*M. pachydermatis* (BIDLMV 20127)	**0.706**	**0.687**	**0.580**	**0.417**	**0.440**	**0.786**	**0.653**	**0.441**	**0.516**	**0.681**	**0.948**	**0.843**	**0.794**	**1**				
*M. pachydermatis* (BIDLMV 21521	**0.735**	**0.717**	**0.641**	**0.395**	**0.338**	**0.822**	**0.791**	**0.237**	**0.464**	**0.584**	**0.900**	**0.731**	**0.770**	**0.926**	**1**			
TPC	**−0.319**	**−0.085**	** −0.517 **	**0.254**	**0.356**	**−0.002**	**−0.104**	**0.366**	**−0.261**	**−0.133**	**0.206**	**−0.101**	**−0.213**	**0.146**	**0.062**	**1**		
TFC	**−** **0.278**	**−0.075**	** −0.507 **	**0.216**	**0.231**	**0.061**	**−0.016**	**0.124**	**−0.324**	**−0.238**	**0.320**	**−0.043**	**−0.140**	**0.269**	**0.136**	**0.908**	**1**	
DPPH	**−** **0.328**	**0.121**	** −0.531 **	**0.230**	**0.316**	**0.003**	**−0.116**	**0.322**	**−0.227**	**−0.125**	**0.224**	**−0.067**	**−0.192**	**0.183**	**−0.029**	**0.993**	**0.931**	**1**

Red colored values highlight the existence of significant correlations between variables.

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
