# Peer review of "Diversity of Monofloral Honey Based on the Antimicrobial and Antioxidant Potential"

_antibiotics, 2022, doi:10.3390/antibiotics11050595_

Round 1

Reviewer 1 Report

    This manuscript (antibiotics-1701472) obtained many data on bacterial inhibition and antioxidant characteristics of four different monofloral honey. The authors found that the monofloral honey has good antibacterial properties. However, there are some grammatical and expressive errors that need to be corrected. Please find my comments:

Minor comments:

  1. Line 42-43: “bacterial mutants resistant to many commonly used antibiotics” changed as “bacterial resistance”.
  2. Line 105-113: The sentence is too long. Please change it.
  3. Line 205-207: The sentence has grammar mistake. Please change it.
  4. Line 252: Please delete “has”.
  5. Line 256: “BH and AH affected the strains of P. aeruginosa starting from 15%”. What is the meaning? Please revise it.
  6. Line 259: “Regarding” change as “Regarding to”.
  7. Line 269: “de growth” change as “the growth”.
  8. Table 1 and Table 2: How to interpret these data? I think the difference between the data calculated by statistical analysis is more indicative of the data.
  9. Line 386: “notable” change as “significant”.
  10. Line 415-417: Please revise “claim” as “claimed”.
  11. Line 418: “has” change as ”had”.
  12. Line 504: “minimum inhibitory concentration” change as “MIC”.

Reviewer 2 Report

In my opinion, this manuscript should be accepted in present form.

Author Response

We would like to address all our gratitude for the time spent for the review and for the recommendation.

Reviewer 3 Report

The reviewed manuscript entitled "Diversity of Monfloral Honey based on the antimicrobial and antioxidant potential”  is an interesting paper, where the authors determine the antioxidant profile and the antimicrobial activity of four different types of monofloral honey (manuka (MH), brassica – rapeseed (BH), acacia (AH) and linden honey (LH)) against some bacterial/fungal ATCC strains and some multidrug-resistant strains isolated from chronic otitis in dogs.  Results indicated the clinical isolates were sensitive to the antibacterial activity of honey.

The introduction provides a good general basis for the topic. Materials and methods are appropriate. The findings and discussion are clearly explained and the conclusions are mostly well supported by the findings. However, English language minor spell check required.

Author Response

We would like to address all our thanks and gratitude for the constructive observations. 

The entire manuscript has been carefully checked and corrected.

This manuscript is a resubmission of an earlier submission. The following is a list of the peer review reports and author responses from that submission.

Round 1

Reviewer 1 Report

 This article (antibiotics-1626267) reported a study covering antimicrobial and antioxidant potency of different types of monofloral honey using the spectrophotometric and microbiological methods and found manuka honey has the most efficacy and antioxidant potency compared with the other kinds of honeys. However, the manuscript is less innovative and poor writing. The confused figures and tables restrict the presentation of the main contents. Thus, I suggest this article should be rejected according to the present version.

Please find my comments below:

  1. Why choose this four different monofloral honey to study?
  2. What is the basis for selecting canine otitis isolated strains for bacteriological evaluation? These kinds of honey can be used in pet clinic in the future?
  3. In all, there are too many paragraphs in the content and the contents should be highly summarized and condensed.

Reviewer 2 Report

Dear authors, congratulations for this study. I find it suitable for suggesting its publication. Results are clearly presented and the discussion could be interesting for readers. I have found some minor errors/inaccuracies in the English, so I suggest a revision in this aspect.

Another suggestion is to put the names of the species in italics in figures 4 onwards (check names on the axes, titles...)

Reviewer 3 Report

The article entitled "Antimicrobial And Antioxidant Potency of Different Types of Monofloral Honey " describes the antioxidant profile and the antimicrobial activity of 4 different types of monofloral honey (manuka (MH), brassica – rapeseed (BH), acacia (AH) and linden honey (LH)) against microbial strains (referent strains and multidrug-resistant strains isolated from chronic otitis in dogs).

I have read carefully the paper and find it perceptive and important for the scientific community. However, certain issues must be answered. In my view, the manuscript needs to be improved.

My questions and suggestions are listed below:

 Title

The title is not attractive. I suggest rewriting “Diversity of Monfloral Honey based on antimicrobial and antioxidant potential” or something similar. This is only a suggestion for upgrading, not a request.

 Introduction

line 90 and similar – avoid using our, we, in our previous work

Please, try to provide more information about the previously published data with the same aim. Also, you need to overview the previous scientific literature that publish selected Monofloral honey samples and highlight the novelty in your work. Therefore, the aim of the study is not clearly presented and needs to be improved.

Material and methods

Define ATCC and BIDLMV in subsection 4.2. Antimicrobial activity

line 583 – Do you use 10-3 dilution prepared after initial tube with 0.5 McFarland, or your 10-3 is approx. McFarland number 0.5? Please, be more precise

the same comment for line 606

Results and discussion – I will comment together

These sections do not provide sufficient explanations, at first for the used methodology, and comparison with other studies. The examination of antimicrobial activity, as one of the aims of this study, has been reduced to basic methods, which is insufficient and needs to be elaborated in much more detail.

 Additionally, where is in Figure 1 “Different letters among samples indicate significant differences (p < 0.05) among values according to the t-test.”.  Check all figures in this view.

Also, for better interpretation of antioxidant and antimicrobial results, I need to required advanced statistical tools. Please provide the obtained results as part of PCA analysis, global sensitive model, etc.

Conclusions

Avoid “our”...

Section Conclusions required further perspective(s) of this study.

Reviewer 4 Report

Manuscript ID: antibiotics-1626267

Title: Antimicrobial And Antioxidant Potency of Different Types of Monofloral Honey

The study covers the topic of antimicrobial activity of four (manuka-MH, acacia-AH, linden-LH and rapeseed(brassica)-BH) types of honey with connection to its chemical properties (total polyphenol content, total flavonoid content and DPPH radical-scavenging). The antimicrobial activity of mentioned types of honey was tested on several strains of bacteria and mycelia. As a result, the authors established the level of antimicrobial activity of tested honeys MH>BH>AH>LH.

The authors limited themselves to presenting the influence of particular types of honey on the antimicrobial activity, but they do not try to explain it on the basis of the chemical composition of the tested honeys, which would significantly increase the value of the publication.

In line 56 the authors state that "Several components contribute to the antimicrobial activity of honey, such as sugar content, endogenous hydrogen peroxide, polyphenol compounds, 1,2-dicarbonyl compounds, and bee defensin-1". If we follow the presented results more closely, it appears that the manuka honey (for which the highest values of total polyphenols, total flavonoids and DPPH radical-scavenging activity were recorded) exhibits the highest antimicrobial activity, whilst linden honey that was second in terms of content of mentioned ingredients (and in case of total flavonoid content, it was significantly ahead of acacia and rapeseed honeys) was classified as the least active. This result may be helpful in pointing the ingredients responsible for the antimicrobial effect of honey components, however, it was not widely undertaken by the authors.

In section 2 Results point 2.2. Antimicrobial Activity, a significant part of the text repeats data available on diagrams. Apart from that in some cases the ranges of values are presented in a way that makes them difficult to read, e.g. line 156 "BIR ranged from 20.01% up to – 32.27%" may suggest that second value is negative, which does not seem to be true in the light of the data presented in the chart.

Please consider rewriting the text with those values to make them more readable:

line 156: "BIR ranged from 20.01% up to – 32.27%" -> "BIR ranged from 20.01% to 32.27%"

line 168: "BIR values between 18.65%-25.42%" -> BIR values between 18.65% and 25.42%

line 154: "from 29.78%-46.07%" -> from 29.78% to 46.07% and many others.

The legends in the charts are incomplete, e.g.:

Figures 4 -7: no marker added for the value of 20%

Figures 11: only two of the five clinical isolates that were tested are marked - similarly in figures 8-10 and others.

As measurements of optical density were performed in triplicate please add statistics for factors describing the growth of microorganisms (e.g. standard deviations in diagrams)

The discussion section in many places looks like "state of the art". In this part, the reader can get acquainted with a long list of results obtained in previous studies (as a result, the manuscript contains 76 references, which may resemble a review paper), but that part the authors do not focus on capturing the main differences and similarities to the results obtained by other researchers - which would be useful in trying to elucidate the mechanisms of the antimicrobial activity of honeys. The authors confine themselves only to stating that "These differences between the MIC values of manuka honey, obtained by various authors, can be justified on the one hand by the different methods and techniques chosen to test the strains, and on the other hand, by the variety of strains tested."

Editorial errors

The work should be checked for editorial errors

line 143: "BGR for MGR" shouldn't be "BGR or MGR" etc.

The work in its form and methodology is similar to the previous studies describing the antimicrobial effect of honey and although it does not attempt to formulate broader conclusions about the mechanism of the antimicrobial action of honeys, due to the wide spectrum of the strains used and the determination of the most important chemical indicators, the presented data may be useful in formulating more general conclusions in the future.